# Effect of Health Education on Schistosomiasis Control Knowledge, Attitude, and Practice after Schistosomiasis Blocking: Results of a Longitudinal Observational Study in the Field

**DOI:** 10.3390/tropicalmed8050267

**Published:** 2023-05-06

**Authors:** Jiaxin Feng, Xinyi Wang, Xia Zhang, Hehua Hu, Jingbo Xue, Chunli Cao, Jing Xu, Pin Yang, Shizhu Li

**Affiliations:** 1National Institute of Parasitic Diseases, Chinese Center for Disease Control and Prevention (Chinese Center for Tropical Diseases Research), NHC Key Laboratory of Parasite and Vector Biology, WHO Collaborating Center for Tropical Diseases, National Center for International Research on Tropical Diseases, Shanghai 200025, China; 2Jiangling Center for Disease Control and Prevention, Jingzhou 434000, China; 3School of Global Health, Chinese Center for Tropical Diseases Research, Shanghai Jiao Tong University School of Medicine, Shanghai 200025, China

**Keywords:** intervention strategies, transmission risk, Chinese experience, schistosomiasis, health education

## Abstract

Objectives: Schistosomiasis is a zoonotic infectious disease that seriously harms people’s physical and mental health. As early as 1985, the WHO suggested that health education and health promotion should be the focus of schistosomiasis prevention work. This study aimed to explore the effect of health education on controlling the risk of schistosomiasis transmission after schistosomiasis blocking and to provide a scientific basis for the further improvement of intervention strategies after schistosomiasis interruption in China and other endemic countries. Methods: In Jiangling County, Hubei Province, China, one severe, one moderate, and one mildly endemic village were selected as the intervention group; two severe, two moderate, and two mildly endemic villages were selected as the control group. In towns with different epidemic types, a primary school was randomly selected for intervention. In September 2020, a baseline survey was carried out through a questionnaire survey to understand the knowledge, attitudes, and practices (KAP) of adults and students concerning schistosomiasis control. Next, two rounds of health education interventions for schistosomiasis control were carried out. The evaluation survey was conducted in September 2021 and the follow-up survey conducted in September 2022. Results: Compared with the baseline survey, the qualified rate of the KAP on schistosomiasis prevention of the control group in the follow-up survey increased from 79.1% (584/738) to 81.0% (493/609) (*p* > 0.05); in the intervention group, the qualified rate of the KAP on schistosomiasis control increased from 74.9% (286/382) to 88.1% (260/295) (*p* < 0.001). In the baseline survey, the qualified rate of the KAP of the intervention group was lower than that of the control group, and in the follow-up survey, the qualified rate of the KAP of the intervention group was 7.2% higher than that of the control group (*p* < 0.05). Compared with the baseline survey, the accuracy rates of the KAP of the intervention group’s adults were higher than those of the control group, with statistical significance (*p* < 0.001). Compared with the baseline survey, the qualified rate of the students’ KAP in the follow-up survey increased from 83.8% (253/302) to 97.8% (304/311) (*p* < 0.001). In the follow-up survey, the accuracy rate of the knowledge, attitudes, and practices of the students was significantly different from the baseline accuracy (*p* < 0.001). Conclusion: a health education-led risk control model of schistosomiasis can significantly improve schistosomiasis control knowledge among adults and students, establishing correct attitudes and leading to the development of correct hygiene habits.

## 1. Introduction

Schistosomiasis is an infectious disease that seriously harms people’s physical and mental health and affects economic and social development [1,2,3]. Humans, cattle, and other mammals are infected by schistosome when they come into contact with water containing schistosome cercariae [4]. There are five main types of schistosomiasis, which are schistosomiasis japonicum, schistosomiasis haematobium, schistosomiasis mansoni, schistosomiasis mekongi, and schistosomiasis intercalate [5]. Schistosomiasis japonicum is mainly distributed in China, the Philippines, Indonesia, and other places, and schistosomiasis mekongi is mainly distributed in Laos and Cambodia, and they are widespread, neglected tropical diseases (NTDs) of public health concern [6]. The latest survey showed that the prevalence of schistosomiasis has decreased significantly in Asian countries and is close to elimination or transmission interruption [7]. In the Philippines, only one province has a human infection rate of more than 5%, 12 provinces fluctuate between 1% and 5%, and 14 provinces have an infection rate of less than 1% [8]. The prevalence of schistosomiasis japonicum among Indonesian residents fluctuates between 0.3% and 4.8% [9]. The infection rate of schistosomiasis among residents in Laos and Cambodia has dropped below 1% [10].

In the past, schistosomiasis was seriously prevalent in China, and it was widespread in 12 provinces and municipalities in the middle and lower reaches of the Yangtze River. At the beginning of its control, 10 million people were affected by schistosomiasis and 100 million people were threatened. After nearly 70 years, schistosomiasis control strategies implemented in China have achieved remarkable results [7,11,12]. In particular, since the beginning of the 21st century, in order to further contain schistosomiasis, China has implemented a comprehensive control strategy focusing on infection source control [13]. By 2015, China had achieved the target of schistosomiasis transmission control [14]. By the end of 2020, five provinces—Shanghai, Zhejiang, Fujian, Guangdong, and Guangxi—have maintained the standards for the elimination of schistosomiasis; moreover, four provinces—Jiangsu, Sichuan, Hubei, and Yunnan—have met the standards for interrupting the transmission of schistosomiasis [15]. By the end of 2021, 439 (97.3%) of the 451 counties in China where schistosomiasis is endemic had reached transmission interruption or elimination [16].

Although the prevalence of schistosomiasis in China has been at a low level, the epidemic factors affecting the transmission of schistosomiasis have not fundamentally changed. For example, there are multiple infectious sources, such as the widespread and complex breeding environment of intermediate host snails [17]. Therefore, the risk of schistosomiasis transmission is still widespread and long term, and the sustained containment and elimination of schistosomiasis remains a long way off [18,19]. Therefore, distilling the Chinese experience related to schistosomiasis control and contributing Chinese wisdom on schistosomiasis control to efforts in Asia and Africa have become the main focus of research at present [20,21]. To explore a sustainable development strategy for schistosomiasis control after transmission interruption, this study selected adults and students in a marshland schistosomiasis epidemic area to be participants in field experiments on health education-led transmission risk control. A longitudinal observational study, through parallel and before and after comparisons, was conducted to provide a scientific basis for the further improvement of intervention strategies after schistosomiasis interruption in China and other endemic countries.

## 2. Methods

### 2.1. Study Site

Jiangling County, Hubei Province, a typical marshland endemic area in the lower-middle reaches of the Yangtze River, was selected as the study area. Jiangling County is located at the central and southern parts of Hubei Province, on the north bank of the Jingjiang River section of the middle reaches of the Yangtze River, which once was a serious endemic area of schistosomiasis in Hubei Province. Jiangling County has a unique geographical environment and climate, numerous lakes, crisscrossed ditches, and frequent precipitation, which are very suitable for snails’ breeding. The county has 11 towns (administrative areas and farms) and 198 administrative villages (subfields and teams), all of which belong to the schistosomiasis endemic areas, and this is the key endemic county for schistosomiasis prevention and control in Hubei Province [22].

### 2.2. Study Population

According to the historical epidemiological data on schistosomiasis in Jiangling County, the endemic villages were classified as severe, moderate, and mild. Villages with a serological positive rate of more than 10% each year over the past five years were regarded as severely endemic villages; villages with a serological positive rate of less than 5% each year were regarded as mildly endemic villages; and the remaining villages were regarded as moderately endemic villages. One endemic village from each classification was selected as an intervention group, while two villages from each classification were selected as the control group (Figure 1). In towns with different epidemic types, a primary school was randomly selected for the intervention. According to pretests and references, *α* = 0.05 and *β* = 0.05 were set, and the qualified rate of the KAP of adults was set as 70% in the control group and 80% in the intervention group, while the qualified rate of the KAP of students was set as 85% before intervention and 95% after intervention. Therefore, the sample size of adults in each group was calculated to be at least 221, and for students it was at least 269. With the village group as the unit, using cluster random sampling at least 100 adults were selected from each village for the survey. All fifth grade students in the primary schools selected from towns with different epidemic types were surveyed, and the number for each school was not fewer than 100. The statistical efficacy (1 − *β*) was 0.95, which is higher than the recommended value of 0.8.

### 2.3. Baseline Survey

Based on the characteristics of schistosomiasis transmission and prevalence, the knowledge, attitudes, and practices (KAP) questionnaire on schistosomiasis control was designed to meet the characteristics of adults and students. The content of the questionnaire involved basic information and schistosomiasis prevention knowledge, attitudes, and practices. In the prevention practice survey questionnaire for adults, there were 20 single-choice questions and 3 multiple-choice questions. The student questionnaire consisted of 19 questions, all of which were single choice. The data collectors were trained on how to conduct the survey. Individuals completed the questionnaire after an explanation by the data collectors, while those who were illiterate or semi-illiterate were helped by the data collectors. Five percent of the interviewed individuals were randomly selected and re-interviewed by the first author. After the pretests, the reliability coefficient of the questionnaire was higher than 0.7 (0.83 and 0.79 in this study, respectively). The formal baseline survey was conducted in September 2020.

### 2.4. Health Education

Two rounds of health education on schistosomiasis control were conducted for the intervention subjects (the first round started in September 2020 after the baseline survey, and the second round started in March 2021). For adults, face-to-face joint lectures on agricultural technology and schistosomiasis control were held for the intervention group (attendance reached 912). A question-and-answer session with prizes was set up, and disposable paper cups and tissue printed with schistosomiasis knowledge were handed out as prizes; easy-to-understand schistosomiasis prevention propaganda videos were broadcast (watched by 918 people); schistosomiasis prevention leaflets were distributed (1635 copies issued); electronic warning devices were installed in ditches with snail habitats near residential areas (3 in total). The control group did not undergo the above intervention. For students, there were five “One Health” educational tasks on schistosomiasis prevention every semester: took a schistosomiasis prevention class, published a schistosomiasis prevention blackboard newspaper, wrote a schistosomiasis prevention composition, took a schistosomiasis prevention knowledge examination, and held a schistosomiasis prevention activities.

### 2.5. Evaluation Survey and Follow-Up Survey

An evaluation survey was conducted in both the control and intervention groups in September 2021, and a follow-up survey was conducted in September 2022. The evaluation survey and follow-up survey were similar to the baseline survey, except that the schistosomiasis control practice was changed to survey the health behaviors of adults and students over the past year.

### 2.6. Statistical Analysis

Data were double entered into Excel (2019, Microsoft, Redmond, WA, USA) by two different researchers and cross-checked for accuracy by the first author. SPSS (version 19, IBM, Armonk, NY, USA) was used for the statistical analysis. Adults were judged to be qualified if they answered 70% (i.e., 14 questions) of the questions correctly, while for students it was 90% (i.e., 17 questions). A percentage (%) is used to describe the correctness of the schistosomiasis control knowledge, attitudes, and practices. The *χ*^2^ test was used to compare the demographic characteristics and changes in KAP before and after intervention between the control and intervention groups. If the number was small, Fisher’s exact probability test was used instead of the *χ*^2^ test. *p* < 0.05 was considered statistically significant.

### 2.7. Ethical Approval and Patients’ Informed Consent

The National Institute of Parasitic Diseases, Chinese Center for Disease Control and Prevention (Chinese Center for Tropical Diseases Research) granted approval for this study (ethics approval number: 2021019). For the questionnaire survey and health education intervention in this study, informed consent was obtained from all respondents themselves and the school leaders.

## 3. Results

### 3.1. Questionnaire Collection and Basic Information of Survey Objects

The returned valid questionnaires were sorted and analyzed, and the results were comparable between the intervention and control groups (Table 1 and Table 2).

### 3.2. Changes in Schistosomiasis Prevention KAP among Adults

#### 3.2.1. Changes in the Qualified Rate of Schistosomiasis Prevention KAP among Adults

Compared with the baseline, the qualified rate of the schistosomiasis prevention KAP of adults in the intervention group increased by 17.2% (*χ*^2^ = 38.696, *p* < 0.001) in the evaluation and increased by 13.3% (*χ*^2^ = 18.772, *p* < 0.001) in the follow-up. In the evaluation, the qualified rate of the KAP among adults in the intervention group was 7.1% higher than that in the control group (*χ*^2^ = 10.841, *p* = 0.001); in the follow-up, the qualified rate in the intervention group was 7.2% higher than that in the control group (*χ*^2^ = 7.370, *p* = 0.007) (Figure 2).

#### 3.2.2. Changes in the Qualified Rate of Schistosomiasis Prevention KAP among Adults with Different Demographic Characteristics

In the intervention group, only for people aged 18–44 years old (*p* = 0.079) and people with a high school education or above (*p* = 1.000), there was no significant difference in the qualified rate of the KAP between the follow-up and baseline (Table 3). In the follow-up, the subjects were male (*χ*^2^ = 5.548, *p* = 0.018), 18–44 years old (*p* = 0.008), 45–59 years old (*χ*^2^ = 7.211, *p* = 0.007), and had a middle school education (*χ*^2^ = 19.053, *p* < 0.001), and the qualified rate of the intervention group was statistically significant compared with the control group (Table 4).

#### 3.2.3. Changes in the Accuracy Rate of Schistosomiasis Prevention KAP

In the follow-up, the accuracy rate of the knowledge, attitudes, and practices among adults in the intervention group was significantly different from that in the baseline (*p* < 0.001). Compared with the baseline, the knowledge accuracy, attitude accuracy, and practice accuracy of the intervention group in the evaluation and follow-up surveys were higher than those of the control group, and the differences were statistically significant (*p* < 0.001) (Table 5).

#### 3.2.4. Changes in the Accuracy Rate of Schistosomiasis Prevention KAP among Adults with Different Demographic Characteristics

Table 6 and Table 7 show the accuracy rate of schistosomiasis prevention KAP among adults with different demographic characteristics in the control group and the intervention group before and after health education. After health education, the practice accuracy rate of adults with different demographic characteristics in the intervention group was significantly different from that of the control group (*p* < 0.05) (Table 8).

#### 3.2.5. Changes in the Accuracy Rate for Different Questions among Adults

In the baseline survey, the accuracy rate of the schistosomiasis prevention KAP among adults was low. The accuracy rate of 25% of the questions was lower than the 50% among adults in the intervention group, and the accuracy rate of some questions was lower than 20%, such as “What are the common symptoms of schistosomiasis?” (19.9%) and “What protective measures did you take when farming or doing activities in areas with snails?” (0.3%). In the evaluation survey, the accuracy rate of most of the questions significantly improved (*p* < 0.05) among adults in the intervention group, but the accuracy rate of some questions was still lower than 60%, such as “What are the common symptoms of schistosomiasis?” (57.8%), “What protective measures did you take when farming or doing activities in areas with snails?” (49.0%), and “Will you cooperate with the department of schistosomiasis control when it eliminate snails? What actions will you take?” (57.8%). In the follow-up, the accuracy rate of 95% of the questions was higher than 60% among adults (Table 9).

### 3.3. Changes in the Schistosomiasis Prevention KAP among Students

#### 3.3.1. Changes in the Qualified Rate of Schistosomiasis Prevention KAP among Students

Compared with baseline, the qualified rate of students’ schistosomiasis prevention KAP increased by 15.6% in the evaluation (*χ*^2^ = 46.657, *p* < 0.001) and 14.0% (*χ*^2^ = 36.045, *p* < 0.001) in the follow-up (Figure 1).

#### 3.3.2. Changes in the Accuracy Rate of Schistosomiasis Prevention KAP among Students

In the follow-up, the accuracy rates of the schistosomiasis prevention knowledge, attitudes, and practices among students were 99.9% (2797/2799), 99.4% (2164/2177), and 99.0% (924/933), respectively, and the difference was statistically significant compared with the baseline (*p* < 0.001) (Table 10).

#### 3.3.3. Changes of Accuracy Rate of Different Questions among Students

In the baseline survey, the accuracy of some questions was less than 90% among students. In follow-up, the accuracy rate of 79.0% questions was significantly improved among students (*p* < 0.001) (Table 11).

## 4. Discussion

In 2012, the WHO first proposed and adopted a resolution on the elimination of schistosomiasis (WHA 65.21). In 2021, “*Ending the Neglect to Attain the Sustainable Development Goals: A Road Map for Neglected Tropical Diseases 2021–2030”* was issued by the WHO, which sets targets for schistosomiasis control at various stages to be met by 2030. It was predicted that in 2023, 63% of the countries worldwide would have eliminated schistosomiasis, 88% of the countries in 2025, and all countries should have eliminated it in 2030. “*The Outline of the Healthy China 2030 Plan”* also specifies that all endemic counties in China will meet the standards for schistosomiasis elimination by 2030. At present, schistosomiasis control in China has reached transmission interruption, but the prevalence of schistosomiasis may be underestimated due to the wide distribution of snails and insufficient sensitivity of existing detection techniques [23,24,25]. In order to achieve the goal of eliminating schistosomiasis by 2030, the task of interrupting schistosomiasis transmission in China remains daunting, and the intervention strategies after schistosomiasis blocking still need to be improved and optimized [26].

Schistosomiasis is mainly caused by human beings or mammals such as cattle and sheep coming into contact with water containing schistosome cercariae. Therefore, schistosome infection is closely related to human and mammal behavior. Takeuchi et al. [27] showed that residents who washed clothes by a lake every day were 3.30 times more likely to be infected with schistosome than those who never washed clothes by the lake. Liu et al. [28] showed that residents with free-grazing livestock were 1.82 times more likely to be infected with schistosome than those with captive livestock. Therefore, understanding the transmission route of schistosomiasis and developing healthy habits can effectively prevent the transmission of schistosomiasis [29]. Health education is an effective and cost-effective way to change health behaviors [30,31]. As early as 1985, the WHO suggested that health education and health promotion should be the focus of schistosomiasis prevention work [32]. Hu et al. [33] conducted schistosomiasis control health education for adult female residents in the schistosomiasis epidemic area of Poyang Lake in China for 17 consecutive years, and the infection rate of schistosomiasis decreased significantly. Dr. Mott, a former official at the WHO, summarized the experience of schistosomiasis prevention in countries around the world and pointed out that, in addition to drug treatment, health education and safe water should be taken as the main intervention measures [34]. “*The Outline of the Healthy China 2030 Plan”* and "*The Regulations on Schistosomiasis Prevention and Control”* issued by China also emphasize the important role of health education in schistosomiasis control. Therefore, in this study, a longitudinal observational study on transmission risk management focused on health education was conducted to optimize intervention strategies after transmission interruption and to share experience in schistosomiasis control in China for Asian and African schistosomiasis.

In this study, various contents and forms of schistosomiasis prevention health education were carried out in the intervention group. The face-to-face joint lectures on schistosomiasis control knowledge and agricultural production and aquaculture technology for residents not only improved their participation and enthusiasm but also led to a change in their views on schistosomiasis control, which became more rational. This health education initiative made full use of multimedia projectors, the mobile phone Wechat platform, Douyin, and other modern scientific and technological means to carry out innovative campaigns so that residents could learn schistosomiasis prevention knowledge more intuitively and achieve twice the result with half the effort. At the same time as the publicity campaign, with the distribution of schistosomiasis prevention educational knowledge, schistosomiasis prevention knowledge was printed on disposable paper cups and paper as gifts to encourage residents to actively participate in the question and answer sessions so that residents would remember the knowledge more deeply. In addition, an alarm device that warns of the presence of snails was installed in ditches near residential areas, which can not only broadcast schistosomiasis control knowledge but also dissuade residents from touching infected water when they approach ditches. After attending the schistosomiasis prevention knowledge course, students should organize schistosomiasis prevention-related activities under the guidance of a teacher, such as creating children’s songs and games, using the input of gained knowledge for the output of knowledge, enhancing the memory of knowledge.

This study shows that after health education, the qualified rate of adults in the intervention group and the students’ schistosomiasis prevention KAP was higher than 85%, which significantly improved compared with the baseline (*p* < 0.001). The survey results of adults showed that the intervention group with people aged 18–44 years old and people with a senior high school education or above, the follow-up survey compared with the baseline survey had no statistical significance in the qualified rate of knowledge, attitudes, and practices (*p* > 0.05). The reason may be that these two groups of people had relatively high schistosomiasis prevention literacy and a high qualified rate of knowledge, attitudes, and practices at the baseline survey. In the evaluation survey, the qualified rate of the schistosomiasis prevention KAP of the intervention group for those who were female, aged 60 years or older, and had an education level of primary school or below was higher than that of the control group (*p* < 0.05), and there was no statistical significance between the intervention group and the control group in the follow-up survey (*p* > 0.05). The results suggest that long-term health education interventions should be strengthened for females, adults aged 60 years or older, and those with education at the primary school level or below.

Many studies show that practices were the hardest to change in the KAP model. The accuracy rates of the schistosomiasis prevention practices of adults and students were lower than for knowledge and attitudes in this study, which is similar to the research results of Wei et al. [35]. In the follow-up survey, 14.9% of the adults were still in direct contact with water without taking any protective measures during cultivation or activities in snail habitats, 10.2% of adults still caught shrimp in the wild, and 10.9% of adults still swam in the wild, suggesting that the effective guidance of healthy behaviors should be strengthened to promote and consolidate adults’ health behaviors. In the evaluation survey, the students’ accuracy rates for three questions on schistosomiasis prevention behavior did not significantly improve, which may be because the accuracy rates of these questions were relatively high before intervention, reaching more than 96%. The schistosomiasis control KAP in the control group also improved, which may be related to the following factors. First, the control group was not a complete blank control but also received routine health education, such as posting schistosomiasis prevention posters and snail situation announcements. Secondly, schistosomiasis prevention personnel carry out schistosomiasis detection and snail control every year, which also publicizes schistosomiasis prevention. Third, some adults sought answers to the questions at baseline. However, the accuracy rates of the schistosomiasis prevention knowledge, attitudes, and practices in the control group were significantly lower than those in the intervention group during evaluation and follow-up (*p* < 0.001), indicating that the health education in the intervention group was more effective.

The results of the study show that the qualified rate of schistosomiasis control KAP decreased in the follow-up, suggesting that schistosomiasis control knowledge will fade with the passage of time [36,37]. In addition, with the gradual decline in schistosomiasis rates in various countries, people’s awareness of schistosomiasis control is gradually receding. Low-cost and cost-effective health education can not only improve people’s compliance with schistosomiasis control work, such as schistosomiasis detection and treatment and snail detection and snail control, but can also effectively prevent people from coming into contact with infected water. Countries where schistosomiasis is endemic should develop targeted long-term schistosomiasis health education programs to lay the foundation for primary schistosomiasis prevention [36]. This study only reflects the short-term changes in the KAP of adults and students after receiving health education. Continuous interventions over many years will highlight the important role of health education in the risk control of schistosomiasis transmission.

## 5. Conclusions

After schistosomiasis blocking, the further control and elimination of schistosomiasis are two important aspects of current research on schistosomiasis. In this study, the Chinese experience in schistosomiasis control after transmission interruption was shared with endemic countries of the world using a longitudinal observational survey. A health education-led risk control model of schistosomiasis can significantly improve schistosomiasis control knowledge among adults and students and develop correct attitudes and better hygiene habits.

## Figures and Tables

**Figure 1 tropicalmed-08-00267-f001:**
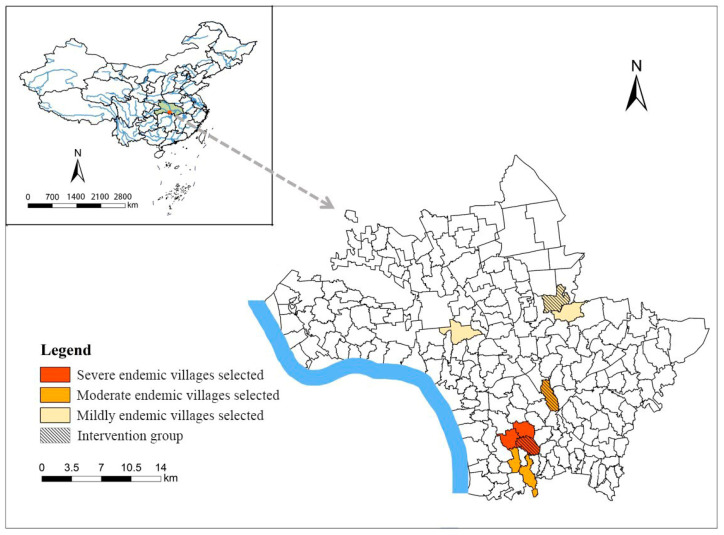
Map of the study site in Jiangling County, Hubei Province, People’s Republic of China.

**Figure 2 tropicalmed-08-00267-f002:**
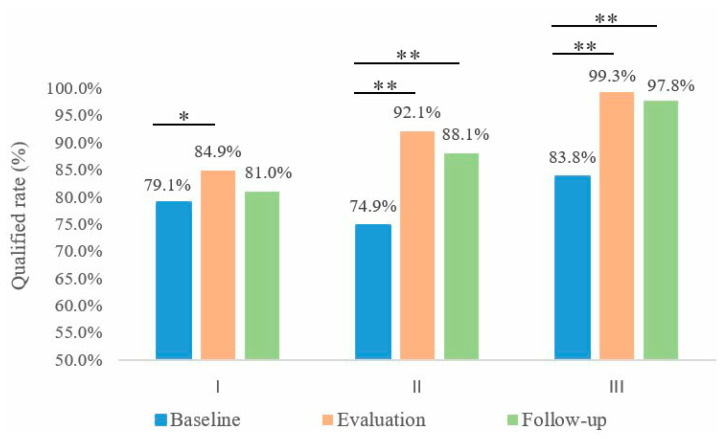
Changes in the qualified rate of the schistosomiasis control KAP among adults and students: (I) qualified rate of the schistosomiasis control KAP among adults in the control group; (II) qualified rate of the schistosomiasis control KAP among adults in the intervention group; (III) qualified rate of the schistosomiasis control KAP among students. Comparison of the evaluation survey and the follow-up survey with the baseline survey: * *p* < 0.05; ** *p* < 0.001. KAP: knowledge, attitudes, and practices.

**Table 1 tropicalmed-08-00267-t001:** The distribution and recovery of questionnaires.

Questionnaires	Adults						Students		
	Baseline		Evaluation		Follow-Up		Baseline	Evaluation	Follow-Up
	Control Group	Intervention Group	Control Group	Intervention Group	Control Group	Intervention Group			
Distribution	748	387	725	359	620	297	302	299	311
Recovery	738	382	717	353	609	295	302	298	311

**Table 2 tropicalmed-08-00267-t002:** Demographic characteristics of subjects in the control group and intervention group.

Characteristic	Baseline				Evaluation				Follow-Up			
	Control Group	Intervention Group	*χ*^2^ Value	*p*-Value	Control Group	Intervention Group	*χ*^2^ Value	*p*-Value	Control Group	Intervention Group	*χ*^2^ Value	*p*-Value
Gender			0.791	0.374			0.780	0.377			0.020	0.889
Male	52.4% (387/738)	55.2% (211/382)			47.8% (343/717)	50.7% (179/353)			53.0% (323/609)	52.5% (155/295)		
Female	47.6% (351/738)	44.76% (171/382)			52.2% (374/717)	49.3% (174/353)			47.0% (286/609)	47.5% (140/295)		
Age, years			5.544	0.063			2.721	0.257			0.545	0.762
18–44	11.4% (84/738)	14.7% (56/382)			10.5% (75/717)	13.3% (47/353)			11.7% (71/609)	12.9% (38/295)		
45–59	43.1% (318/738)	46.6% (178/382)			39.3% (282/717)	40.8% (144/353)			40.6% (247/609)	41.7% (123/295)		
≥60	45.5% (336/738)	38.7% (148/382)			50.2% (360/717)	45.9% (162/353)			47.8% (291/609)	45.4% (134/295)		
Education			5.329	0.07			4.739	0.094			5.591	0.061
Primary school and below	57.2% (422/738)	52.9% (202/382)			67.4% (483/717)	75.6% (257/353)			54.0% (329/609)	60.7% (179/295)		
Junior middle school	35.9% (265/738)	36.4% (139/382)			30.5% (219/717)	21.5% (86/353)			39.9% (243/609)	31.9% (94/295)		
Senior high school or above	6.9% (51/738)	10.7% (41/382)			2.1% (15/717)	2.8% (10/353)			6.1% (37/609)	7.5% (22/295)		

**Table 3 tropicalmed-08-00267-t003:** Changes in the qualified rate of schistosomiasis control KAP among adults with different demographic characteristics.

Characteristic	Control Group							Intervention Group						
	Qualified Rate at Baseline (%)	Qualified Rate at Evaluation (%)	Difference in Qualified Rate (%)	Test	Qualified Rate at Follow-Up (%)	Difference in Qualified Rate (%)	Test	Qualified Rate at Baseline (%)	Qualified Rate at Evaluation (%)	Difference in Qualified Rate (%)	Test	Qualified Rate at Follow-Up (%)	Difference in Qualified Rate (%)	Test
Gender														
Male	81.7 (316/387)	85.7 (294/343)	4.1	*χ*^2^ = 2.183, *p* = 0.140	82.0 (265/323)	0.4	*χ*^2^ = 0.018, *p* = 0.893	76.8 (162/211)	93.3 (167/179)	16.5	*χ*^2^ = 20.028, ***p* < 0.001**	90.3 (140/155)	13.5	*χ*^2^ = 11.363, ***p* = 0.001**
Female	76.4 (268/351)	84.0 (315/374)	7.6	*χ*^2^ = 7.123, ***p* = 0.008**	79.7 (228/286)	3.4	*χ*^2^ = 1.307, *p* = 0.309	72.5 (124/171)	90.8 (158/174)	18.3	*χ*^2^ = 19.329, ***p* < 0.001**	85.7 (120/140)	13.2	*χ*^2^ = 7.935, ***p* = 0.005**
Age, years														
18–44	75.0 (63/84)	90.7 (68/75)	15.7	*χ*^2^ = 6.703, ***p* = 0.010**	84.5 (60/71)	9.5	*χ*^2^ = 2.123, *p* = 0.145	91.1 (51/56)	97.9 (46/47)	6.8	*p* = 0.216	100 (38/38)	8.9	*p* = 0.079
45–59	82.7 (263/318)	89.4 (252/282)	6.7	*χ*^2^ = 5.448, ***p* = 0.020**	87.0 (215/247)	4.3	*χ*^2^ = 6.020, ***p* = 0.014**	77.5 (138/178)	94.4 (136/144)	16.9	*χ*^2^ = 17.958, ***p* < 0.001**	95.9 (118/123)	18.4	*χ*^2^ = 19.382, ***p* < 0.001**
≥60	76.8 (258/336)	80.3 (289/360)	3.5	*χ*^2^ = 1.260, *p* = 0.262	74.9 (218/291)	1.9	*χ*^2^ = 0.299, *p* = 0.585	65.5 (97/148)	88.3 (143/162)	22.7	*χ*^2^ = 22.860, ***p* < 0.001**	77.6 (104/134)	12.1	*χ*^2^ = 5.006, ***p* = 0.025**
Education														
Primary school and below	73.9 (312/422)	82.0 (396/483)	8.1	*χ*^2^ = 8.579, ***p* = 0.003**	80.6 (265/329)	6.6	*χ*^2^ = 4.542, ***p* = 0.033**	69.3 (140/202)	90.7 (233/257)	21.4	*χ*^2^ = 33.874, ***p* < 0.001**	81.6 (146/179)	12.3	*χ*^2^ = 7.618, ***p* = 0.006**
Junior middle school	84.9 (225/265)	91.3 (200/219)	6.4	*χ*^2^ = 4.615, ***p* = 0.032**	80.3 (195/243)	4.7	*χ*^2^ = 1.921, *p* = 0.166	77.7 (108/139)	95.4 (82/86)	17.7	*χ*^2^ = 12.601, ***p* < 0.001**	98.9 (93/94)	21.2	*χ*^2^ = 21.350, ***p* < 0.001**
Senior high school or above	92.2 (47/51)	86.7 (13/15)	5.5	*p* = 0.612	89.2 (33/37)	3.0	*p* = 0.716	92.7 (38/41)	100 (10/10)	7.3	*p* = 1.000	95.5 (21/22)	2.8	*p* = 1.000

The “difference in the qualified rate” was compared with the baseline qualified rate of each group. *p*-Values highlighted in bold mean the difference was statistically significant. KAP: knowledge, attitudes, and practices.

**Table 4 tropicalmed-08-00267-t004:** Comparison of the qualified rate of schistosomiasis control KAP among adults with different demographic characteristics between the control group and the intervention group after health education.

Characteristics	Difference in Qualified Rate at Evaluation (%)	Test of Difference at Evaluation	Difference in Qualified Rate at Follow-Up (%)	Test of Difference at Follow-Up
Gender				
Male	7.6	*χ*^2^ = 6.551, ***p* = 0.010**	8.3	*χ*^2^ = 5.548, ***p* = 0.018**
Female	6.8	*χ*^2^ = 4.352, ***p* = 0.037**	6.0	*χ*^2^ = 2.258, *p* = 0.133
Age, years				
18–44	7.2	*p* = 0.151	15.5	***p* = 0.008**
45–59	5.1	*χ*^2^ = 3.031, *p* = 0.082	8.9	*χ*^2^ = 7.211, ***p* = 0.007**
≥60	8.0	*χ*^2^ = 5.003, ***p* = 0.025**	2.7	*χ*^2^ = 0.364, *p* = 0.546
Education				
Primary school and below	8.7	*χ*^2^ = 9.898, ***p* = 0.002**	1.0	*χ*^2^ = 0.078, *p* = 0.781
Junior middle school	4.0	*χ*^2^ = 1.435, *p* = 0.231	18.7	*χ*^2^ = 19.053, ***p* < 0.001**
Senior high school or above	13.3	*p* = 0.500	6.3	*p* = 0.641

*p*-Values highlighted in bold mean the difference was statistically significant. KAP: knowledge, attitudes, and practices.

**Table 5 tropicalmed-08-00267-t005:** Changes in the accuracy rate of schistosomiasis control KAP among adults.

Type	Control Group							Intervention Group							Difference in Accuracy Rate at Evaluation (%)	Test of Difference at Evaluation	Difference in accuracy rate at Follow-Up (%)	Test of difference in Follow-Up
	Accuracy Rate at Baseline (%)	Accuracy Rate at Evaluation (%)	Difference in Accuracy Rate (%)	Test	Accuracy Rate at Follow-Up (%)	Difference in Accuracy Rate (%)	Test	Accuracy Rate at Baseline (%)	Accuracy Rate at Evaluation (%)	Difference in Accuracy Rate (%)	Test	Accuracy Rate at Follow-Up (%)	Difference in Accuracy Rate (%)	Test				
Knowledge	78.3 (4622/5904)	84.6 (4855/5736)	6.4	*χ*^2^ = 77.662, *p* < 0.001	89.8 (4375/4872)	11.5	*χ*^2^ = 256.691, *p* < 0.001	79.0 (2414/3056)	88.5 (2500/2824)	9.5	*χ*^2^ = 97.186, *p* < 0.001	93.7 (2212/2360)	14.7	*χ*^2^ = 232.119, *p* < 0.001	3.9	*χ*^2^ = 23.626, *p* < 0.001	3.9	*χ*^2^ = 30.228, *p* < 0.001
Attitude	83.0 (3674/4428)	86.8 (3734/4302)	3.8	*χ*^2^ = 24.842, *p* < 0.001	87.8 (3207/3654)	4.8	*χ*^2^ = 36.379, *p* < 0.001	76.6 (1755/2292)	91.6 (1940/2118)	15.0	*χ*^2^ = 182.935, *p* < 0.001	91.9 (1626/1770)	15.3	*χ*^2^ = 167.403, *p* < 0.001	4.8	*χ*^2^ = 31.828, *p* < 0.001	4.1	*χ*^2^ = 20.621, *p* < 0.001
Practice	56.6 (2505/4428)	60.7 (2609/4302)	4.1	*χ*^2^ = 14.929, *p* < 0.001	60.0 (2194/3654)	3.5	*χ*^2^ = 9.916, *p* = 0.002	53.2 (1219/2292)	80.6 (1706/2118)	27.4	*χ*^2^ = 369.015, *p* < 0.001	70.5 (1247/1770)	17.3	*χ*^2^ = 124.835, *p* < 0.001	19.9	*χ*^2^ = 255.076, *p* < 0.001	10.4	*χ*^2^ = 55.693, *p* < 0.001

The “difference in accuracy rate” was compared with the baseline accuracy rate of each group. Accuracy rate = number of accurate people and questions/total number of people and question × 100%. KAP: knowledge, attitudes, and practices.

**Table 6 tropicalmed-08-00267-t006:** Changes in the accuracy rate of schistosomiasis control KAP among adults with different demographic characteristics in the control group.

Characteristic	Accuracy Rate at Baseline (%)			Accuracy Rate at Evaluation (%)			Accuracy Rate at Follow-Up (%)		
	Knowledge	Attitude	Practice	Knowledge	Attitude	Practice	Knowledge	Attitude	Practice
Gender									
Male	78.8 (2441/3096)	83.8 (1945/2322)	56.4 (1310/2322)	85.1 (2336/2744)	87.2 (1794/2058)	61.7 (1270/2058)	90.5 (2339/2584)	87.8 (1702/1938)	61.0 (1182/1938)
Female	77.7 (2181/2808)	82.1 (1729/2106)	56.7 (1195/2106)	84.2 (2519/2992)	86.5 (1940/2244)	59.7 (1339/2244)	89.0 (2036/2288)	87.7 (1505/1716)	59.0 (1012/1716)
Age									
18~44	72.8 (489/672)	84.7 (427/504)	56.0 (282/504)	86.8 (521/600)	88.7 (399/450)	64.0 (288/450)	94.4 (536/568)	87.1 (371/426)	56.1 (239/426)
45~59	79.9 (2032/2544)	83.7 (1596/1908)	56.9 (1085/1908)	87.0 (1963/2256)	87.9 (1488/1692)	62.1 (1050/1692)	89.8 (1774/1976)	88.1 (1305/1482)	61.5 (911/1482)
≥60	78.2 (2101/2688)	81.9 (1651/2016)	56.5 (1138/2016)	82.3 (2371/2880)	85.5 (1847/2160)	58.8 (1271/2160)	88.7 (2065/2328)	87.7 (1531/1746)	59.8 (1044/1746)
Education									
Primary school and below	77.1 (2603/3376)	81.0 (2052/2532)	54.7 (1384/2532)	83.4 (3224/3864)	85.9 (2489/2898)	59.5 (1723/2898)	86.9 (2286/2632)	88.7 (1750/1974)	60.7 (1199/1974)
Junior middle school	79.3 (1681/2120)	85.0 (1351/1590)	59.1 (940/1590)	87.0 (1525/1752)	89.0 (1170/1314)	63.0 (828/1314)	92.9 (1805/1944)	87.5 (1275/1458)	59.5 (868/1458)
Senior high school or above	82.8 (338/408)	88.6 (271/306)	59.2 (181/306)	88.3 (106/120)	83.3 (75/90)	64.4 (58/90)	96.0 (284/296)	82.0 (182/222)	57.2 (127/222)

KAP: knowledge, attitudes, and practices.

**Table 7 tropicalmed-08-00267-t007:** Changes in the accuracy rate of schistosomiasis control KAP among adults with different demographic characteristics in the intervention group.

Characteristic	Accuracy Rate at Baseline (%)			Accuracy Rate at Evaluation (%)			Accuracy Rate at Follow-Up (%)		
	Knowledge	Attitude	Practice	Knowledge	Attitude	Practice	Knowledge	Attitude	Practice
Gender									
Male	80.0 (1351/1688)	78.2 (990/1266)	52.3 (662/1266)	89.9 (1287/1432)	93.8 (1007/1074)	81.8 (879/1074)	94.4 (1170/1240)	93.4 (869/930)	70.0 (651/930)
Female	77.7 (1063/1368)	74.6 (765/1026)	54.3 (557/1026)	87.1 (1213/1392)	88.8 (927/1044)	79.2 (827/1044)	93.0 (1042/1120)	90.1 (757/840)	71.0 (596/840)
Age									
18~44	83.0 (372/448)	86.0 (289/336)	56.9 (191/336)	94.4 (355/376)	95.0 (268/282)	83.3 (235/282)	94.7 (288/304)	98.7 (225/228)	75.0 (171/228)
45~59	79.4 (1130/1424)	78.9 (843/1068)	54.1 (578/1068)	91.3 (1052/1152)	94.7 (818/864)	83.0 (717/864)	95.2 (937/984)	96.1 (709/738)	74.5 (550/738)
≥60	77.0 (912/1184)	70.2 (623/888)	50.7 (450/888)	84.3 (1093/1296)	87.9 (854/972)	77.6 (754/972)	92.1 (987/1072)	86.1 (692/804)	65.4 (526/804)
Education									
Primary school and below	75.4 (1219/1616)	73.8 (894/1212)	50.4 (611/1212)	86.7 (1782/2056)	90.2 (1391/1542)	78.8 (1215/1542)	92.1 (1319/1432)	88.2 (947/1074)	67.6 (726/1074)
Junior middle school	81.0 (901/1112)	76.7 (640/834)	53.6 (447/834)	93.5 (643/688)	95.4 (492/516)	84.5 (436/516)	96.0 (722/752)	98.1 (553/564)	74.7 (421/564)
Senior high school or above	89.6 (294/328)	89.8 (221/246)	65.5 (161/246)	93.8 (75/80)	95.0 (57/60)	91.7 (55/60)	97.2 (171/176)	95.5 (126/132)	75.8 (100/132)

KAP: knowledge, attitudes, and practices.

**Table 8 tropicalmed-08-00267-t008:** Comparison of the accuracy rate of schistosomiasis control KAP among adults with different demographic characteristics between the control group and the intervention group after health education.

Characteristic	Knowledge				Attitude				Practice			
	Difference in Accuracy Rate at Evaluation (%)	Test of Difference at Evaluation	Difference in Accuracy Rate at Follow-Up (%)	Test of Difference at Follow-Up	Difference in Accuracy Rate at Evaluation (%)	Test of Difference at Evaluation	Difference in Accuracy Rate at Follow-Up (%)	Test of Difference at Follow-Up	Difference in Accuracy Rate at Evaluation (%)	Test of Difference at Evaluation	Difference in Accuracy Rate at Follow-Up (%)	Test of Difference at Follow-Up
Gender												
Male	4.7	*χ*^2^ = 18.426, ***p* < 0.001**	3.8	*χ*^2^ = 16.314, ***p* < 0.001**	6.6	*χ*^2^ = 32.423, ***p* < 0.001**	5.6	*χ*^2^ = 21.369, ***p* < 0.001**	20.1	χ^2^ = 132.833, ***p* < 0.001**	9.0	*χ*^2^ = 22.115, ***p* < 0.001**
Female	3.0	*χ*^2^ = 6.529, ***p* = 0.011**	4.1	*χ*^2^ = 14.101, ***p* < 0.001**	2.3	*χ*^2^ = 3.495, *p* = 0.062	2.42	*χ*^2^ = 3.231, *p* = 0.072	19.5	χ^2^ = 121.072, ***p* < 0.001**	12.0	*χ*^2^ = 34.676, ***p* < 0.001**
Age, years												
18–44	7.6	*χ*^2^ = 14.448, ***p* < 0.001**	0.4	*χ*^2^ = 0.052, *p* = 0.819	6.4	*χ*^2^ = 8.691, ***p* = 0.003**	11.6	*χ*^2^ = 24.705, ***p* < 0.001**	19.3	χ^2^ = 31.764, ***p* < 0.001**	18.9	*χ*^2^ = 22.674, ***p* < 0.001**
45–59	4.3	*χ*^2^ = 13.867, ***p* < 0.001**	5.4	*χ*^2^ = 25.289, ***p* < 0.001**	6.7	*χ*^2^ = 29.380, ***p* < 0.001**	8.0	*χ*^2^ = 37.584, ***p* < 0.001**	20.9	χ^2^ = 117.402, ***p* < 0.001**	13.1	*χ*^2^ = 37.317, ***p* < 0.001**
≥60	2.0	*χ*^2^ = 2.553, *p* = 0.110	3.4	*χ*^2^ = 9.063, ***p* = 0.003**	2.4	*χ*^2^ = 3.122, *p* = 0.077	1.6	*χ*^2^ = 1.287, *p* = 0.257	18.7	χ^2^ = 102.901, ***p* < 0.001**	5.6	*χ*^2^ = 7.372, ***p* = 0.007**
Education												
Primary school and below	3.2	*χ*^2^ = 10.766, ***p* = 0.001**	5.3	*χ*^2^ = 25.561, ***p* < 0.001**	4.3	*χ*^2^ = 17.047, ***p* < 0.001**	0.5	*χ*^2^ = 0.156, *p* = 0.693	19.3	χ^2^ = 168.155, ***p* < 0.001**	6.9	*χ*^2^ = 14.060, ***p* < 0.001**
Junior middle school	6.4	*χ*^2^ = 20.531, ***p* < 0.001**	3.2	*χ*^2^ = 9.220, ***p* = 0.002**	6.3	*χ*^2^ = 17.681, ***p* < 0.001**	10.6	*χ*^2^ = 52.691, ***p* < 0.001**	21.5	χ^2^ = 80.039, ***p* < 0.001**	15.1	*χ*^2^ = 40.188, ***p* < 0.001**
Senior high school or above	5.4	*χ*^2^ = 1.638, *p* = 0.201	1.2	*χ*^2^ = 0.468, *p* = 0.494	11.7	*χ*^2^ = 4.640, ***p* = 0.031**	13.5	*χ*^2^ = 13.290, ***p* < 0.001**	27.2	χ^2^ = 14.357, ***p* < 0.001**	18.6	*χ*^2^ = 12.382, ***p* < 0.001**

*p*-Values highlighted in bold mean the difference was statistically significant. KAP: knowledge, attitudes, and practices.

**Table 9 tropicalmed-08-00267-t009:** Changes in the accuracy rate of each question of schistosomiasis control KAP among adults.

Question	Control Group							Intervention Group						
	Accuracy Rate at Baseline (%)	Accuracy Rate at Evaluation (%)	Difference in Accuracy Rate (%)	Test	Accuracy Rate at Follow-Up (%)	Difference in Accuracy Rate (%)	Test	Accuracy Rate at Baseline (%)	Accuracy Rate at Evaluation (%)	Difference in Accuracy Rate (%)	Test	Accuracy Rate at Follow-Up (%)	Difference in Accuracy Rate (%)	Test
1. Are mammals such as cattle and sheep susceptible to schistosomiasis?	82.4 (608/738)	89.3 (640/717)	6.9	*χ*^2^ = 14.090, ***p* < 0.001**	97.5 (594/609)	15.2	*χ*^2^ = 79.747, ***p* < 0.001**	84.8 (324/382)	93.2 (329/353)	8.4	*χ*^2^ = 13.012, ***p* < 0.001**	99.7 (294/295)	14.8	*χ*^2^ = 46.105, ***p* < 0.001**
2. Is Schistosomiasis transmitted by human and animal feces?	78.2 (577/738)	82.6 (592/717)	4.4	*χ*^2^ = 4.422, ***p* = 0.035**	96.1 (585/609)	17.9	*χ*^2^ = 89.980, ***p* < 0.001**	84.0 (321/382)	89.2 (315/353)	5.2	*χ*^2^ = 4.262, ***p* = 0.039**	99.0 (292/295)	15.0	*χ*^2^ = 43.472, ***p* < 0.001**
3. Which animal is the intermediate host of schistosomiasis?	95.1 (702/738)	94.6 (678/717)	0.6	*χ*^2^ = 0.234, *p* = 0.628	96.1 (585/609)	0.9	*χ*^2^ = 0.689, *p* = 0.407	94.8 (362/382)	95.8 (338/353)	1.0	*χ*^2^ = 0.394, *p* = 0.530	94.2 (278/295)	0.5	*χ*^2^ = 0.090, *p* = 0.765
4. What causes schistosomiasis?	88.6 (654/738)	89.3 (640/717)	0.6	*χ*^2^ = 0.153, *p* = 0.696	96.4 (587/609)	7.8	*χ*^2^ = 27.782, ***p* < 0.001**	82.2 (314/382)	93.8 (331/353)	11.6	*χ*^2^ = 22.851, ***p* < 0.001**	98.0 (289/295)	15.8	*χ*^2^ = 42.504, ***p* < 0.001**
5. When is the most susceptible time of year for schistosomiasis?	89.8 (663/738)	93.0 (667/717)	3.2	*χ*^2^ = 4.710, ***p* = 0.030**	94.6 (576/609)	4.7	*χ*^2^ = 10.182, ***p* = 0.001**	89.3 (341/382)	95.2 (336/353)	5.9	*χ*^2^ = 8.838, ***p* = 0.003**	99.3 (293/295)	10.1	*χ*^2^ = 28.293, ***p* < 0.001**
6. What are the common symptoms of schistosomiasis?	13.6 (100/738)	48.1 (345/717)	34.6	*χ*^2^ = 204.683, ***p* < 0.001**	56.2 (342/609)	42.6	*χ*^2^ = 274.753, ***p* < 0.001**	19.9 (76/382)	57.8 (204/353)	37.9	*χ*^2^ = 111.7183, ***p* < 0.001**	77.6 (229/295)	57.7	*χ*^2^ = 224.110, ***p* < 0.001**
7. What is the simplest and most effective way to prevent schistosomiasis?	91.6 (676/738)	89.3 (640/717)	2.3	*χ*^2^ = 2.301, *p* = 0.129	96.2 (586/609)	4.6	*χ*^2^ = 12.069, ***p* = 0.001**	89.3 (341/382)	90.9 (321/353)	1.7	*χ*^2^ = 0.570, *p* = 0.450	94.6 (279/295)	5.3	*χ*^2^ = 6.085, ***p* = 0.014**
8. After recovering from schistosomiasis, can you get it again?	87.0 (642/738)	91.1 (653/717)	4.1	*χ*^2^ = 6.192, ***p* = 0.013**	85.4 (520/609)	1.6	*χ*^2^ = 0.726, *p* = 0.394	87.7 (335/382)	92.4 (326/353)	4.7	*χ*^2^ = 4.391, ***p* = 0.036**	87.5 (258/295)	0.2	*χ*^2^ = 0.009, *p* = 0.926
9. What is your attitude when the department of schistosomiasis control calls for screening for schistosomiasis?	32.7 (241/738)	65.1 (467/717)	32.5	*χ*^2^ = 153.542, ***p* < 0.001**	94.3 (574/609)	61.6	*χ*^2^ = 529.775, ***p* < 0.001**	36.9 (141/382)	81.3 (287/353)	44.4	*χ*^2^ = 148.647, ***p* < 0.001**	80.3 (237/295)	43.4	*χ*^2^ = 127.306, ***p* < 0.001**
10. When you see a place with a schistosomiasis warning sign, what do you do?	74.1 (547/738)	78.5 (563/717)	4.4	*χ*^2^ = 3.896, ***p* = 0.048**	93.4 (569/609)	19.3	*χ*^2^ = 87.589, ***p* < 0.001**	60.5 (231/382)	84.7 (299/353)	24.2	*χ*^2^ = 53.561, ***p* < 0.001**	84.8 (250/295)	24.3	*χ*^2^ = 47.684, ***p* < 0.001**
11. Do you want to learn about schistosomiasis control?	96.6 (713/738)	94.0 (674/717)	2.6	*χ*^2^ = 5.559, *p* = 0.018	77.5 (472/609)	19.1	*χ*^2^ = 141.257, *p* < 0.001	84.3 (322/382)	96.0 (339/353)	11.7	*χ*^2^ = 27.931, ***p* < 0.001**	86.1 (254/295)	1.8	*χ*^2^ = 0.429, *p* = 0.513
12. Which ways do you like to learn about schistosomiasis prevention?	100 (738/738)	94.0 (674/717)	6.0	*χ*^2^ = 45.607, *p* < 0.001	99.8 (594/609)	0.2	*χ*^2^ = 18.382, *p* < 0.001	94.8 (362/382)	96.0 (339/353)	1.3	*χ*^2^ = 0.670, *p* = 0.413	100.0 (295/295)	5.2	*χ*^2^ = 15.915, ***p* < 0.001**
13. Do you want to know the current status of native snail distribution and disease detection?	94.4 (697/738)	94.6 (678/717)	0.1	*χ*^2^ = 0.009, *p* = 0.923	82.1 (500/609)	12.3	*χ*^2^ = 51.366, *p* < 0.001	88.0 (336/382)	95.8 (338/353)	7.8	*χ*^2^ = 14.639, ***p* < 0.001**	100.0 (295/295)	12.0	*χ*^2^ = 38.113, ***p* < 0.001**
14. Which ways do you like to learn about native snail distribution and disease detection?	100 (738/738)	94.6 (678/717)	5.4	*χ*^2^ = 41.248, *p* < 0.001	79.5 (484/609)	20.5	*χ*^2^ = 166.973, *p* < 0.001	95.0 (363/382)	95.8 (338/353)	0.7	*χ*^2^ = 0.218, *p* = 0.640	100.0 (295/295)	5.0	*χ*^2^ = 15.096, ***p* < 0.001**
15. Do you cooperate when testing for schistosomiasis infection?	93.2 (688/738)	94.7 (679/717)	1.5	*χ*^2^ = 1.393, *p* = 0.238	89.8 (547/609)	3.4	*χ*^2^ = 5.076, *p* = 0.024	96.1 (367/382)	96.3 (340/353)	0.3	*χ*^2^ = 0.030, *p* = 0.863	95.9 (283/295)	0.1	*χ*^2^ = 0.009, *p* = 0.926
16. Had you ever fished or shrimped in the wild?	78.1 (576/738)	70.0 (502/717)	8.0	*χ*^2^ = 12.230, *p* < 0.001	63.7 (388/609)	14.3	*χ*^2^ = 33.708, *p* < 0.001	65.5 (250/382)	86.1 (304/353)	20.7	*χ*^2^ = 42.246, ***p* < 0.001**	89.8 (265/295)	24.4	*χ*^2^ = 54.377, ***p* < 0.001**
17. Did you go swimming in the wild in the summer?	95.9 (708/738)	92.5 (663/717)	3.5	*χ*^2^ = 8.033, *p* = 0.005	77.7 (473/609)	17.8	*χ*^2^ = 103.039, *p* < 0.001	89.3 (341/382)	98.0 (346/353)	8.8	*χ*^2^ = 23.011, ***p* < 0.001**	89.2 (263/295)	0.1	*χ*^2^ = 0.002, *p* = 0.962
18. What protective measures did you take when farming or doing activities in areas with snails?	0.7 (5/738)	4.5 (32/717)	3.8	*χ*^2^ = 21.029, ***p* < 0.001**	4.1 (25/609)	3.4	*χ*^2^ = 18.002, ***p* < 0.001**	0.3 (1/382)	49.0 (173/353)	48.8	*χ*^2^ = 241.270, ***p* < 0.001**	0 (0/295)	0.3	*p* = 1.000
19. What did you do with your untreated feces?	57.6 (425/738)	72.9 (523/717)	15.4	*χ*^2^ = 37.767, ***p* < 0.001**	93.4 (569/609)	35.8	*χ*^2^ = 221.670, ***p* < 0.001**	47.6 (182/382)	96.0 (339/353)	48.4	*χ*^2^ = 208.154, ***p* < 0.001**	85.1 (251/295)	37.4	*χ*^2^ = 101.225, ***p* < 0.001**
20. Will you cooperate with the department of schistosomiasis control when it eliminates snails? What actions will you take?	14.0 (103/738)	29.3 (210/717)	15.3	*χ*^2^ = 50.632, ***p* < 0.001**	31.5 (192/609)	17.6	*χ*^2^ = 60.224, ***p* < 0.001**	20.4 (78/382)	57.8 (204/353)	37.4	*χ*^2^ = 108.358, ***p* < 0.001**	62.7 (185/295)	42.3	*χ*^2^ = 125.330, ***p* < 0.001**

The “difference in accuracy rate” was compared with the baseline accuracy rate of each group. *p*-Values highlighted in bold mean higher than the baseline accuracy rate, and the difference was statistically significant. KAP: knowledge, attitudes, and practices.

**Table 10 tropicalmed-08-00267-t010:** Changes in the accuracy rate of schistosomiasis control KAP among students.

Type	Accuracy Rate at Baseline (%)	Accuracy Rate at Evaluation (%)	Difference in Accuracy Rate (%)	Test	Accuracy Rate at Follow-Up (%)	Difference in Accuracy Rate (%)	Test
Knowledge	91.8 (2494/2718)	99.6 (2672/2682)	7.9	*χ*^2^ = 201.612, *p* < 0.001	99.9 (2797/2799)	8.2	*χ*^2^ = 234.284, *p* < 0.001
Attitude	93.7 (1981/2114)	99.5 (2075/2086)	5.8	*χ*^2^ = 105.358, *p* < 0.001	99.4 (2164/2177)	5.7	*χ*^2^ = 105.807, *p* < 0.001
Practice	96.7 (876/906)	98.2 (878/894)	1.5	*χ*^2^ = 4.183, *p* = 0.041	99.0 (924/933)	2.4	*χ*^2^ = 12.194, *p* < 0.001

The “difference in accuracy rate” was compared with the baseline accuracy rate of each group. Accuracy rate = number of accurate people and questions/total number of people and question × 100%. KAP: knowledge, attitudes, and practices.

**Table 11 tropicalmed-08-00267-t011:** Changes in the accuracy rate of each question of schistosomiasis control KAP among students.

Question	Accuracy Rate at Baseline (%)	Accuracy Rate at Evaluation (%)	Difference in Accuracy Rate (%)	Test	Accuracy Rate at Follow-Up (%)	Difference in Accuracy Rate (%)	Test
1. Do you know about schistosomiasis?	96.4 (291/302)	99.3 (296/298)	3.0	*χ*^2^ = 6.247, ***p* = 0.012**	100.0 (311/311)	3.6	*χ*^2^ = 11.535, ***p* = 0.001**
2. Which aquatic animals transmit schistosomiasis?	75.8 (229/302)	99.0 (295/298)	23.2	*χ*^2^ = 72.763, ***p* < 0.001**	100.0 (311/311)	24.2	*χ*^2^ = 85.338, ***p* < 0.001**
3. Do cattle transmit schistosomiasis?	85.4 (258/302)	99.3 (296/298)	13.9	*χ*^2^ = 40.929, ***p* < 0.001**	100.0 (311/311)	14.6	*χ*^2^ = 48.815, ***p* < 0.001**
4. What are the common symptoms of schistosomiasis patients?	95.0 (287/302)	99.7 (297/298)	4.6	*χ*^2^ = 12.395, ***p* < 0.001**	100.0 (311/311)	5.0	*χ*^2^ = 15.834, ***p* < 0.001**
5. What causes schistosomiasis?	95.7 (289/302)	99.7 (297/298)	4.0	*χ*^2^ = 10.369, ***p* = 0.001**	100.0 (311/311)	4.3	*χ*^2^ = 13.677, ***p* < 0.001**
6. Is schistosomiasis harmful to human health?	95.0 (287/302)	100 (298/298)	5.0	*χ*^2^ = 15.181, ***p* < 0.001**	99.7 (310/311)	4.7	*χ*^2^ = 13.007, ***p* < 0.001**
7. What is a symptom of advanced schistosomiasis patient?	94.7 (286/302)	100 (298/298)	5.3	*χ*^2^ = 16.221, ***p* < 0.001**	100.0 (311/311)	5.3	*χ*^2^ = 16.918, ***p* < 0.001**
8. What is a symptom of children getting schistosomiasis?	97.4 (294/302)	100 (298/298)	2.7	***p* = 0.007**	100.0 (311/311)	2.7	***p =* 0.003**
9. Which of the following animals gets schistosomiasis?	90.4 (273/302)	99.7 (297/298)	9.3	*χ*^2^ = 27.118, ***p* < 0.001**	99.7 (310/311)	9.3	*χ*^2^ = 28.356, ***p* < 0.001**
10. Do you go to the river to catch fish or shrimp?	97.0 (293/302)	99.7 (297/298)	2.6	***p* = 0.020**	97.8 (304/311)	0.7	*χ*^2^ = 0.321, *p* = 0.571
11. Where do you usually swim in summer?	99.0 (299/302)	100 (298/298)	1.0	*p* = 0.249	100.0 (311/311)	1.0	*p =* 0.119
12. Have you ever taken a health education course on schistosomiasis?	91.4 (276/302)	99.7 (297/298)	8.3	*χ*^2^ = 23.892, ***p* < 0.001**	100.0 (311/311)	8.6	*χ*^2^ = 27.961, ***p* < 0.001**
13. Would you discourage other students from playing in the water near the ditch?	96.7 (292/302)	100 (298/298)	3.3	***p* = 0.002**	98.4 (306/311)	1.7	*χ*^2^ = 1.863, *p* = 0.172
14. What do you do when the department of schistosomiasis control is testing for schistosomiasis infection?	94.7 (286/302)	99.7 (297/298)	5.0	*χ*^2^ = 13.417, ***p* < 0.001**	99.7 (310/311)	5.0	*χ*^2^ = 14.073, ***p* < 0.001**
15. What is the best way to prevent schistosomiasis?	84.1 (254/302)	97.3 (290/298)	13.2	*χ*^2^ = 30.928, ***p* < 0.001**	100.0 (311/311)	15.9	*χ*^2^ = 53.630, ***p* < 0.001**
16. Can Schistosomiasis be prevented?	93.1 (281/302)	100 (298/298)	7.0	*χ*^2^ = 21.473, ***p* < 0.001**	100.0 (311/311)	7.0	*χ*^2^ = 22.393, ***p* < 0.001**
17. Have you ever been infected and treated for schistosomiasis?	97.0 (293/302)	98.7 (294/298)	1.6	*χ*^2^ = 1.898, *p* = 0.168	100.0 (311/311)	3.0	***p =* 0.002**
18. Have you swam or caught fish in the river last year?	96.0 (290/302)	97.3 (290/298)	1.3	*χ*^2^ = 0.773, *p* = 0.379	97.1 (302/311)	1.1	*χ*^2^ = 0.540, *p* = 0.463
19. Have you ever defecated indiscriminately?	97.0 (293/302)	98.7 (294/298)	1.6	*χ*^2^ = 1.898, *p* = 0.168	100.0 (311/311)	3.0	***p =* 0.002**

*p*-Values highlighted in bold mean the difference was statistically significant. KAP: knowledge, attitudes, and practices.

## Data Availability

Data are not available due to the fact of ethical restrictions.

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
