# Peer review of "Effect of Health Education on Schistosomiasis Control Knowledge, Attitude, and Practice after Schistosomiasis Blocking: Results of a Longitudinal Observational Study in the Field"

_tropicalmed, 2023, doi:10.3390/tropicalmed8050267_

Round 1
Reviewer 1 Report
The authors should show clearly the sample size determination for each of the groups. The authors should show justification for the classification of communities as severe, moderate, and mild. Is there any reference for this categorization?
Author Response
1.The authors should show clearly the sample size determination for each of the groups.
Thank you for your suggestion. We added the relevant details of the sample size calculation. Explain as follows: We calculated that at least 221 residents were surveyed in each group, and since we selected three intervention villages, at least 100 people were surveyed in each village. We calculated that there were at least 269 students surveyed, and since we chose three schools, we surveyed at least 100 students in each primary school.
2.The authors should show justification for the classification of communities as severe, moderate, and mild. Is there any reference for this categorization?
Thank you for your suggestion. We added the reference for this categorization.

Reviewer 2 Report
General observations:
Survey studies are important to assess a parasite epidemiology and evaluate the success rate of intervention in control and prevention strategies. It would have been very valuable to perform a diagnostic test for schistosomiasis prior and after the health education intervention.
Manuscript should be revised by a native English speaker. Revise the use of comma in your sentences. Furthermore, many of the sentences are very long and confusing.
Abstract:
Before the objectives, authors should include a small introduction highlighting the importance of Schistosomiasis and the importance or impact that health education has as a parasite control strategy.
KAP: Acronyms should be described prior to use.
Keywords: schistosomiasis; health education; intervention; field study… As a suggestion, keyword should not be included in the title, because the purpose of keywords is to ensure databases and search engines to quickly identify and show your manuscript to users looking for related topics. Thus, repeating words from the title would lower the probability of your work being found.
Line 47: remove parenthesis (S.) japonicum, moreover, use italics for scientific names.
Line 48: A sentence should never start with abbreviations, S. japonicum, at the beginning of a sentence always full names should be included, please check the manuscript to correct this issue.
Line 53: has an human infection rate- remove the “n”.
Line 56: The infection rate among residents of schistosomiasis in Laos and Cambodia…. This sentence is grammatically incorrect as residents do not live in schistosomiasis, correct? Suggestion: The infection rate of schistosomiasis among residents in Laos and Cambodia….
Line 58: schistosomiasis japonicum. Schistosomiasis is the term referring to the illness caused by S. japanicum, I suggest correcting leaving just schistosomiasis or S. japanicum.
Line 62: Grammar suggestion: After nearly 70 years, schistosomiasis control strategies implemented in China have achieved remarkable results.
Line 63-66: In particular,…. This sentence is too long and has punctuation mistakes, I suggest rewriting.
Line 66: Revise the use of comma in your sentences.
Lines 69-71: Please revise punctuation. Furthermore, when stating that the number of cases was reduced to 29,000. I strongly believe that the original number of cases should be mentioned before, to give a clear sense of the reduction and to let readers know if it was significant.
Lines 73-78. This sentence is also too long and because of that has punctuation mistakes. I suggest rewriting.
Lines 78-81. Confusing sentence, suggest rewriting.
Line 81: To explore a sustainable…
Lines 81-87. This sentence is also too long, I suggest splitting in two sentences.
Line 94: Yangtze River, which once was a serious…
Line 96: Jiangling County has a unique geographical environment and climate, numerous lakes, crisscrossed ditches, and frequent precipitation, which are very suitable for snails breeding.
Line 101-104. I suggest rewriting as follows: According to the historical epidemic data of schistosomiasis in Jiangling County, the endemic villages were classified as: severe, moderate, and mild. One of each endemic villages selected as intervention groups, while two villages from each classification were selected as control group (Figure 1).
Line 119: Correct: In the prevention practice survey questionnaire for adults, there were 20 single choice questions and three multiple-choice.
Line 124: A sentence should never begin with a number. Furthermore, a simple rule for using numbers in writing is that small numbers ranging from one to nine, should be spelled out, while larger numbers (above ten) are written as numerals.
Line 132: were held in the intervention group…
Line 355: at various stages by 2030: By 2023, 63% of… suggestion: …at various stages by 2030. It was predicted that in 2023, 63% of the countries worldwide would have eliminated schistosomiasis, 88% of the countries in 2025, and all countries should have eliminated it in 2030.
Line 374: Health education is an effective and cost-effective way to change health behaviors.
Line 392: residents’… remove apostrophe.
Line 401: an alarm device that warns of snails exist has been set up in ditches near residential areas, which can not only broadcast schistosomiasis control knowledge, but also dissuade residents from touching infected water when they approach ditches.… Suggestion: an alarm device that warns of the presence of snails has been installed in ditches near residential areas; which can not only broadcast schistosomiasis control knowledge, but also dissuade residents from touching infected water when they approach ditches.
Line 430: and consolidate adults’ health behavior… the students’……adult health behavior …and remove apostrophe in students.
Line 446, 447: people is plural… remove the “s” and the apostrophe.
Line 461: Chinese experience in schistosomiasis control after transmission interruption was shared with the world’s endemic countries through longitudinal observation: Health education-led risk control model of schistosomiasis can significantly improve schistosomiasis control knowledge among adults and students, establish correct attitude and develop correct hygiene habits. … suggestion: Chinese experience in schistosomiasis control after transmission interruption was shared through a longitudinal observation survey, with endemic countries of the world: Health education-led risk control model of schistosomiasis can significantly improve schistosomiasis control knowledge among adults and students, to develop a correct attitude and better hygiene habits…
Author Response
Thank you for your suggestion. We have revised all the suggestions you have given.

Reviewer 3 Report
The manuscript is a significant contribution to the effort to eradicate schistosomiasis in China.
I have just a few suggestions:
Line 22 - the knowledge, attitude, and practice … – please add the abbreviation “KAP” – abbreviation should be explained on the first occurrence in the text.
Line 46 - “There are five main types of schistosomiasis, which are schistosomiasis (S.) japonicum, S. haematobium, S. mansoni, S. mekongi and S. intercalate” – schistosomiasis is the name of the disease, not the parasite, it should be: “There are five main types of schistosomiasis, which are Schistosoma (S.) japonicum…”
(S.) japonicum, S. haematobium, S. mansoni, S. mekongi - name of the species should be in italic, please correct in the whole manuscript.
Line 101 - According to the historical epidemic data of schistosomiasis in Jiangling County, the endemic villages were classified as severe, moderate and mildly – these terms are too vague for a scientific article, please specify (express in numbers or other comparable way) what „severe, moderate and mildly“ mean.
Line 214: â…¢: qualified rate of schistosomiasis control KAP among students. – please explain, why are the students not divided into „control group“ and “ intervention group“ as adults.
Author Response
1. Line 22 - the knowledge, attitude, and practice … – please add the abbreviation “KAP” – abbreviation should be explained on the first occurrence in the text.
Thank you for your suggestion. We have modified it according to your suggestion.
2. Line 46 - “There are five main types of schistosomiasis, which are schistosomiasis (S.) japonicum, S. haematobium, S. mansoni, S. mekongi and S. intercalate” – schistosomiasis is the name of the disease, not the parasite, it should be: “There are five main types of schistosomiasis, which are Schistosoma (S.) japonicum…”
(S.) japonicum, S. haematobium, S. mansoni, S. mekongi - name of the species should be in italic, please correct in the whole manuscript.
Thank you for your suggestion. We have modified it according to your suggestion.
3. Line 101 - According to the historical epidemic data of schistosomiasis in Jiangling County, the endemic villages were classified as severe, moderate and mildly – these terms are too vague for a scientific article, please specify (express in numbers or other comparable way) what „severe, moderate and mildly“ mean.
Thank you for your suggestion. We have supplemented the classification criteria according to your suggestion.
4. Line 214: â…¢: qualified rate of schistosomiasis control KAP among students. – please explain, why are the students not divided into „control group“ and “ intervention group“ as adults.
Thank you for your suggestion. Explain as follows: We have set intervention group and control group among the residents, which can achieve the purpose of comparative analysis in this study. So for the students, we did all the interventions, and of course we wanted the students to teach their parents about schistosomiasis control.
